# Implementation processes in a cognitive rehabilitation intervention for people with dementia: a complexity-informed qualitative analysis

Sarah Morgan-Trimmer ![ORCID],[1] Aleksandra Kudlicka,[1] Krystal Warmoth ![ORCID],[2] Iracema Leroi,[3] Jan R Oyebode,[4] Jackie Pool,[5] Robert Woods,[6] Linda Clare[7]

For numbered affiliations see end of article.

**Correspondence to**
Dr Sarah Morgan-Trimmer;
S.Morgan-Trimmer@exeter.ac.uk

## ABSTRACT

**Objectives** Healthcare is often delivered through complex interventions. Understanding how to implement these successfully is important for optimising services. This article demonstrates how the complexity theory concept of 'self-organisation' can inform implementation, drawing on a process evaluation within a randomised controlled trial of the GREAT (*G*oal-oriented cognitive *R*ehabilitation in *E*arly-stage *A*lzheimer's and related dementias: a multi-centre single-blind randomised controlled *T*rial) intervention which compared a cognitive rehabilitation intervention for people with dementia with usual treatment.

**Design** A process evaluation examined experiences of GREAT therapists and participants receiving the intervention, through thematic analysis of a focus group with therapists and interviews with participants and their carers. Therapy records of participants receiving the intervention were also analysed using adapted framework analysis. Analysis adopted a critical realist perspective and a deductive-inductive approach to identify patterns in how the intervention operated.

**Setting** The GREAT intervention was delivered through home visits by therapists, in eight regions in the UK.

**Participants** Six therapists took part in a focus group, interviews were conducted with 25 participants and 26 carers, and therapy logs for 50 participants were analysed.

**Intervention** A 16-week cognitive rehabilitation programme for people with mild-to-moderate dementia.

**Results** 'Self-organisation' of the intervention occurred through adaptations made by therapists. Adaptations included simplifying the intervention for people with greater cognitive impairment, and extending it to meet additional needs. Relational work by therapists produced an emergent outcome of 'social support'. Self-organised aspects of the intervention were less visible than formal components, but were important aspects of how it operated during the trial. This understanding can help to inform future implementation.

**Conclusions** Researchers are increasingly adopting complexity theory to understand interventions. This study extends the application of complexity theory by demonstrating how 'self-organisation' was a useful concept for understanding aspects of the intervention that would have been missed by focusing on formal intervention components. Analysis of self-organisation

## STRENGTHS AND LIMITATIONS OF THIS STUDY

⇒ This study adopted an in-depth, theoretically informed qualitative analysis to understand implementation processes in a complex intervention.

⇒ The final stage of analysis drew on three complementary datasets to develop a holistic understanding of how a complex intervention operated.

⇒ A limitation of the study is that it only reports therapist and participant perceptions of how the intervention operated; researcher observations of therapy sessions were not conducted.

⇒ This study included a small number of focus group participants, interviewees and therapy records; the study was designed to produce theoretical generalisability through in-depth analysis rather than a large sample size.

could enhance future process evaluations and implementation studies.

**Trial registration number** ISRCTN21027481.

## BACKGROUND

### Introduction

This article discusses the application of the complexity theory concept of 'self-organisation' for understanding implementation. Implementation, defined here as 'the process through which interventions are delivered, and what is delivered in practice' (Moore *et al*, p8),[1] is an important element of process evaluation. This is distinct from the definition of implementation within the field of 'implementation science' which 'is the scientific study of methods to promote the systematic uptake of research findings and other evidence-based practices into routine practice' (Eccles and Mittman, p1).[2] While there is overlap between the two definitions, process evaluations tend to focus on 'the quality and quantity of what is actually delivered during the evaluation (Moore *et al*, p10)'[1] and it is this focus we adopt in this

article. It reports findings from a process evaluation which addressed several research questions, including 'how was the intervention implemented?'. Process evaluations may be the first opportunity to closely study, refine and optimise implementation of a new intervention. Findings on implementation can then inform wider roll-out of interventions at the post-trial stage.[1]

Existing approaches to implementation employed by process evaluations face two main limitations. First, process evaluations are often structured using logic models which represent the underlying theory of how an intervention, including implementation processes, work.[3] Logic models usefully provide a simple-enough model on which to base an evaluation, but in doing so they tend to conceptualise the world in a relatively linear, mechanistic way. Second, logic models focus attention on whether a certain process happened, rather than unexpected events, because they represent ideal versions of what researchers hypothesise will occur. Within this approach, implementation of the intervention tends to be examined in terms of fidelity to this ideal version. However, during implementation, some components of the intervention may not have been delivered as anticipated, additional components may have been introduced and components may have interacted with each other or contextual factors in unexpected ways. These aspects may be missed if data are only collected on components identified in logic models. This article discusses the application of the complexity theory concept of 'self-organisation' for improving understanding of how complex health interventions are implemented. It examines how it can be applied through analysing process evaluation data from a trial of a community-based cognitive rehabilitation (CR) intervention for people with dementia.[4 5]

## Complexity theory and self-organisation

Complexity theory has been increasingly advocated as an alternative theoretical lens for understanding complex interventions, and is being increasingly applied across a range of healthcare research areas and evaluation studies.[6–12] It theorises systems of different types—social, physical, biological—as constantly self-organising and unfolding. A complex system cannot be understood only in terms of its parts, as a complicated machine, for example, where the causal powers of the system are explained by its components.[6 9 13] Processes occur in complex systems through the dynamic, non-linear relations between components at different levels as interacting parts of a system, producing effects in unpredictable ways.[14–16] Complexity theory encompasses a range of concepts which describe how patterns occur, including self-organisation, feedback loops and emergence.[10 14 15 17] Complex health interventions can be conceptualised as complex systems since they incorporate multiple interacting elements such as humans (as active and responsive agents), organisations, materials, rules, policies and so on, through which these patterns may occur.[6 13 14] Complexity approaches also conceptualise

the system within which an intervention is introduced as complex, unfolding, interacting with and changed by the intervention: systems are nested within other systems and cannot be isolated.[6 15 17–19]

This article draws on the complexity concept of 'self-organisation', and the additional concepts of 'emergence' and 'feedback loops', and explores their application to studying implementation. 'Self-organisation' refers to interacting elements of a system, such as practitioners, who behave or make individual microlevel decisions in response to other elements of a system (such as clients) in a way which produces patterns or order at a higher level.[10 15] This order is produced in a bottom-up way since it occurs as the result of individual decisions which are not coordinated, rather than through top-down instructions. The order which is created is 'emergent', as it appears at the overall intervention level and occurs somewhat unpredictably. A common form of self-organisation occurs through feedback loops, for example, where practitioners tailor how they deliver an intervention in response to client reactions to the intervention.[13]

Self-organisation can operate within or beyond the formal design of an intervention. Some interventions are designed to be partly self-organising, where the practitioner is expected to adapt the intervention according to individual client needs, for example, as long as the core functions of the intervention are delivered.[20 21] The core functions relate to ways in which the intervention activities produce mechanisms of action, and must be retained for an intervention to be effective, while the activities (form) of an intervention may be adapted.[22] Other forms of self-organisation occur outside of the prescribed parameters of the intervention, through deviation from protocols for example. Through microlevel decisions about adaptations, practitioner behaviours result in self-organisation as their patterns of behaviour at a collective level shape what intervention is actually delivered. Self-organisation is not commonly theorised as being part of implementation processes, yet could significantly influence implementation and, in turn, intervention outcomes.

While complexity theory may explain the real world more adequately than linear models, there are challenges in applying it within a process evaluation. Complexity theory is a collection of related concepts which are often presented in overlapping rather than standardised lists, and does not have an agreed definition.[10] It is not yet a consolidated theory in that there are questions about how concepts are related to each other.[9 17] Complexity theory also tends to emphasise the unpredictability of outcomes and produce post hoc explanations rather than having strong predictive power.[10 23 24] This is somewhat at odds with the aim of process evaluation, and research more generally, to identify generalisable patterns, 'demiregularities' or predictive theories.[6] However, the emphasis on unpredictability varies within the family of complexity theory approaches.[6] This article examines the application of complexity concepts, particularly 'self-organisation', in a process evaluation and reflects on its

use for understanding the implementation of complex health interventions.

The paper draws on several datasets which answered a set of related questions for the process evaluation about how the intervention operated. The research questions were: how was the intervention delivered; what was the feasibility of delivering the intervention; was the intervention delivered with fidelity; what were the mechanisms of impact; what influenced treatment outcomes; what were participant experiences of the intervention; and what were therapists' experiences of delivering the intervention? The analysis reported here answered a further research question: how did the way the intervention was delivered influence whether and how participants benefited? The broad aim of a process evaluation is to explain how an intervention operates to produce its outcomes, but process evaluations often only address different elements of an intervention separately. This approach, drawing on complexity concepts, was an attempt to conduct a more holistic analysis.

### The GREAT trial

The GREAT study (*G*oal-oriented cognitive *R*ehabilitation in *E*arly-stage *A*lzheimer's and related dementias: a multi-centre single-blind randomised controlled *T*rial) was a randomised controlled trial, with an embedded process evaluation, of a CR intervention for people with a clinical diagnosis of dementia (referred to here as 'participants') and a Mini Mental State Examination (MMSE) score of ≥18 points indicating mild to moderate cognitive impairment.[25 26] It was delivered in eight regions in the UK. The main purpose of the intervention was to improve ability to carry out everyday activities in the areas the participant chose to target. The intervention involved identifying with the participant personally relevant and significant goals related to daily activities, and then working together to develop and implement a set of strategies to enable the person to achieve the desired outcomes. The primary outcome was participant-reported progress towards participant-identified goals at 3 months. It was delivered by a practitioner, eight of whom were occupational therapists (OTs) and one a nurse during 1-hour home visits, ideally with a carer also present at part of the session. The participant, with carer support, was encouraged to implement agreed changes in daily routines and practise strategies between visits. Ten visits were delivered over 3 months, followed by four maintenance sessions over 6 months. The intervention was detailed in a practitioner handbook which included a structured protocol[27]; the core component of sessions was to work towards personalised goals, supplemented by strategies to improve attention and concentration, compensatory strategies, and restorative strategies for retaining new information or improving recall. Additional optional components included anxiety management, increasing activity levels, discussing carer well-being and signposting to other services. Training sessions and regular group and individual supervision were provided for the therapists.

The intervention is conceptualised here as a complex intervention as it had multiple interacting components, addressed difficulties encountered in dementia for both participants and carers, was a tailored intervention and targeted multiple outcomes.[28] Full details of the intervention, trial methods and trial outcomes have been published.[4 5 25 29]

The intervention was personalised in that participants could identify up to three goals they wished to work towards and the therapists applied CR strategies to address these. The intervention was also designed to be responsive to participant needs in that therapists had some flexibility to apply elements of the intervention such as anxiety management, depending on the needs of the participant and their personal and social contexts. The handbook reflected this balance between structure and flexibility, for example:

> The intervention in a trial needs to follow a structured protocol, as summarised in table 1 below. However, some flexibility will be needed as participants will have varying needs and preferences and will progress at different rates.
>
> GREAT Handbook for Therapists (Clare and Kudlicka, p39)[27]

This article examines the implementation of the GREAT intervention, where the intervention is conceptualised as a complex intervention designed to incorporate a degree of self-organisation through its person-centred design and requirement to be responsive to the contexts, needs and preferences of people with dementia.

### METHODS

This article draws on findings from three datasets which were analysed for the GREAT trial process evaluation.[4] Methods and findings are reported in accordance with the Standards for Reporting Qualitative Research checklist (online supplemental appendix 1). A focus group and analysis of therapy logs from the intervention examined the perceptions and experiences of therapists about how the intervention was delivered, feasibility of the intervention, fidelity to the intervention protocol and perceived factors affecting treatment outcomes. A set of interviews explored the perceptions and experiences of the intervention by participants and their carers, and whether and how any impact from the intervention was experienced by this group. Written informed consent was obtained at the beginning of the trial. Trial researchers and therapists were trained to monitor ongoing consent and identify and respond to any indication of possible withdrawal of consent. Researchers began participant and carer interviews by re-establishing consent. A critical realist perspective was adopted in the data analysis to identify causal patterns of how the intervention operated (a realist ontology) while allowing that perceptions of the intervention may differ between participants (a

relativist epistemology). To this end, the three datasets were initially analysed with an overall focus on participant perspectives, whether these were categorised according to deductive categories identified from previous research or inductively developed. Additional interpretive work was conducted using a graphic to relate themes from the focus group analysis to each other, to draw out a more realist explanation of the dynamic relationships between different elements of the intervention. In a second stage, a realist-oriented analysis of how the intervention operated was conducted though deductively applying constructs from the complexity literature to the initial findings from the three datasets.

## Focus group

A focus group was conducted with all six therapists who were in post at the time, at the end of the first year of the intervention, to examine their experiences of the intervention (see online supplemental appendix 2 for schedule). The focus group was conducted as part of the second annual training event for the trial therapists. The discussion was facilitated by coauthor and trial coinvestigator JRO. She had a thorough understanding of the intervention but was not involved in supervising the therapists, which facilitated an open discussion. It was digitally recorded, transcribed verbatim and uploaded to NVivo V.11. A thematic analysis, underpinned by a critical realist perspective, was conducted by the lead author, SM-T, who had not been involved in the study up to this point.[30] Data were initially coded using a combined inductive–deductive approach, drawing on the dementia literature informing the GREAT trial.[4] Themes were then developed, after first summarising the codes, and graphics were developed to represent and analyse themes at a more interpretive level. This article draws on three themes relevant to implementation: the perceived influence of the severity of dementia; adaptation work of therapists in response to their perceptions of the severity of dementia; and the relational work implemented by therapists and the outcomes this produced. These themes are more fully described in the findings section.

## Analysis of therapy logs

Therapy logs were maintained for every participant for each of the 14 sessions and included detail about each intervention component, such as progress towards goals (see online supplemental appendix 3 for categories included in therapy logs). These logs were analysed by SMT to try to identify factors which might explain how participants benefited from the intervention, or not. Therapy logs were analysed to compare the 25 participants with the highest-score primary outcomes (the 'good outcomes' group) and the 25 participants with the lowest-score primary outcomes (the 'poor outcomes' group), out of the intervention arm population of 281 individuals. The primary outcome was participant-reported goal attainment at 3 months post-randomisation, measured using the Bangor Goal-Setting

Interview.[4] This measurement was undertaken through a home visit by a researcher who was blinded to the trial arm of the participant.

An adapted framework analysis method was employed, using deductive categories identified in the focus group findings and also inductive categories where novel factors were identified.[31] The analysis aimed to identify differences in patterns in each group, paying attention to both qualitative and quantitative differences. First, each component for each session was summarised and compared between the 'good outcomes' and 'poor outcomes' groups. Second, the trajectory of each participant's progress through the intervention was summarised and then compared by group. Third, a 'negative case analysis' was conducted to explore factors which did not fit with the general patterns emerging from the first two stages of the analysis.

## Interviews

Face-to-face, semi-structured interviews were conducted with 25 participants and 26 carers at 9 months post-intervention, to explore their experiences of the intervention (see online supplemental appendix 4 for interview schedule). They were consecutively sampled across three different sites (North Wales, South Wales and Greater Manchester) where the intervention was taking place and where there was a research assistant not otherwise involved in the trial available to complete the interviews. Participants and carers were interviewed separately wherever possible, starting with the person with dementia. Interviewers took a photograph of the therapist on the visit to prompt the participant's memory of the therapy sessions. If the participant was struggling to recall the therapy sessions, the interview was completed jointly with the carer. All interviews were audiorecorded, transcribed verbatim and uploaded to NVivo V.11.[4]

Data were analysed using thematic analysis which was underpinned by a critical realist position and which employed an inductive approach to identifying and exploring patterns of meaning.[32 33] Data were initially coded by four researchers, and then organised into meaningful groups by KW and SM-T. Related themes were clustered together and organised into an overall thematic map.

Detailed methods and findings of the focus group, therapy log analysis and interviews have been reported.[4] In a last stage of analysis, themes from the focus group analysis were interpreted in relation to core concepts in the complexity literature. The core concepts considered were: interacting elements, unpredictability, self-organisation, emergence, non-linearity, fuzzy boundaries, feedback loops and being 'more than the sum of its parts'.[9 10 14 15 17] These themes were then further developed through triangulation with findings from the interviews and therapy log analysis. Findings on intervention implementation are reported below.

## Patient and public involvement

For the GREAT trial, in which the process evaluation was embedded, experts by experience including Alzheimer's Society Research Volunteers were consulted at the setup stage to inform participant information resources and trial procedures. As a result we made a number of amendments to the participant-facing documents and assessment measures. The inclusion of qualitative interviews and a focus group in the study design was at the suggestion of the experts by experience. They also provided insightful comments about the progress of the trial and contributed to developing a follow-up application for an implementation grant. Study participants were updated about trial progress through a regular newsletter.

## RESULTS

Findings presented here explain how the intervention operated as a complex system, and how some aspects of implementation occurred in a self-organising pattern, through adaptation behaviours of therapists. These were: simplifying the intervention for people with greater cognitive impairment; providing additional support in response to participant needs; and conducting relational work to engage participants, which produced an emergent outcome of increased social support.

### Self-organisation through adaptive response

The GREAT intervention incorporated self-organisation in its design: therapists were required to select CR strategies appropriate for individuals and their social context, and tailor the intervention to individuals' needs and preferences. Participants and carers reported experiencing the intervention as personalised:

> [Goals] were always relevant to … obviously relevant to the issues that [therapist] wanted to raise … And also relevant to, the issues that were important for [person with dementia] … she worked at a pace that was good for him as well.
>
> (Interview, Carer 5)

Therapists reported that they adapted the intervention in response to their perception that participants with greater cognitive impairment (within the trial population range) were less likely to engage in and benefit from the intervention. For example, therapists observed that these participants had more difficulty setting relevant goals or remembering them. Therapists' notes for participants in the 'poor outcomes' group also described features which could be attributable to greater cognitive impairment: a tendency to set more basic-level goals; being more likely to give up on a goal; lower levels of motivation; being more withdrawn during sessions; and having less awareness regarding their condition.

Therapists responded to participants with greater cognitive impairment through adapting their delivery of the intervention for these participants in several ways. First, because it was difficult for these participants to absorb information, therapists would slow down the pace of delivery and only deliver what they thought were the most relevant sections:

> Therapist 5: I was almost going at a snail's pace, because I realised it was far too much information to dish out, and I was literally cherry-picking the bits I thought were relevant so I could get through the goals and get them rated, and just literally [Several people agreeing] Therapist 4: And, sometimes you do have to cherry pick don't you? [Several people agreeing].
>
> (Therapist focus group)

Therapists also tailored the timing of material, moving some supplementary topics to an alternative session, for example:

> Therapist 6: But there's too much information for me, so let alone someone who has to take a while, or so I've started moving things away
>
> Therapist 4: Oh, I've moved things very early on, I moved the anxiety stuff, unless it's the glaring, big problem, I move anxiety from week 1 to week 2 because it's too much information.
>
> (Therapist focus group)

In addition, therapists reported that they simplified some of the language in the handbook:

> Therapist 7: The 'restorative' and 'compensatory', that's too jargon-y, and too heavy for the person, not all, but some of the people with dementia…
>
> Therapist 2: Yes, I've had to change the words to 'methods', because 'strategies' just scares them.
>
> (Therapist focus group)

In these ways, therapists made individual, microlevel decisions to ensure the intervention matched the needs of participants with greater cognitive impairment, and sometimes adapted the content from the intervention protocol.[27] This is an example of self-organisation operating in a feedback loop, where practitioners adapted an intervention according to the perceived capacity of people with dementia. Through this adaptation work, they shaped the nature of intervention for some participants.

### Self-organisation through extending the intervention

Carers were important for the delivery of the intervention because they supported the practice of new techniques, and could also impact on participant motivation. Difficulty engaging some carers was noted in the therapy logs and, in the focus group, therapists reported additional efforts they made to engage carers. For example, therapists sometimes adjusted the time allocated (often increasing it) for home visits and also changed the timing of visits to ensure they met with carers:

> Therapist 7: I mean I've done home visits at 8 o'clock in the evening, just to catch up with the carer, so I

can have face-to-face and actually get what's going on, because I can't get hold of her during the day, she just doesn't answer her phone. So, having to do home visits in the evening, so that's impacting, that's where the hours are coming in, from doing more of that work…

Therapist 6: I've often gone out on home visits, and they've [the carer has] gone out. Then you have to extend your visit by an extra 20, 30 minutes because, you don't want to just leave… So you have to extend your visit sometimes, or you have to ring [the carer] afterwards.

(Therapist focus group)

Therapists sometimes also had to take additional time to identify and meet additional carers who were most likely to be the person supporting the intervention but not the carer who was originally nominated to take part in the intervention:

Therapist 4: Actually the workload increases for us, because what we're doing is we're talking to the carer that's on the spot, and we're also having to liaise with the son and daughter, either by text or email, or something else in the evenings, that has quite a massive implication [agreement from others].

(Therapist focus group)

Therapists also delivered several 'add-on' components in response to the needs of participants and carers. First, the intervention component to address carers' levels of well-being was formally limited in the handbook to referring carers to local sources of support: *You can direct carers to appropriate sources of support in the local area, and encourage them to access these (Clare and Kudlicka, p51).*[27] However, one therapist reported contacting services directly:

Therapist 7: I'm also contacting social services, so that the carer will get a break, that shouldn't be part of my role, but no-one else seems to.

(Therapist focus group)

Second, therapists commented on (often marital) conflict in the relationships between participants and their carers, also noted in therapy logs, and attempted to reduce conflict even though they were aware it was beyond the parameters of the intervention:

Therapist 2: I do find that it's about the nature of the relationship as well. And often you do find, like you're doing a couples intervention, it's not just about the dementia, it's often about the dynamics that have probably gone on through their whole relationship but the situation is highlighting it, and that's really quite difficult to manage, isn't it?

Therapist 6: Sometimes it's like marriage counselling

Therapist 2: Yeah… it's about their relationship, isn't it, it is a bit of the couples stuff, and I find myself

doing that, and I think it is helpful, but it's beyond what we've been given.

(Therapist focus group)

Several carers described in interviews how the therapist acted as a mediator in conflicts between the carer and person with dementia. Carers also reported improved relationships between themselves and participants as a result of the intervention, for example having increased understanding of and patience with the person with dementia. Several participants commented that they had greater social awareness and were more likely to consider the impact of what they said on others.

In these ways, self-organisation occurred through therapists' micro-level decisions to adjust (often increase) the time taken to engage carers, directly contact services, and reduce conflict between carers and people with dementia. Some therapists extended the intervention so that it was larger in scope than described in the handbook. These patterns of self-organisation can also be understood in term of feedback loops, where therapists' adaptive behaviours were in response to the contexts and unmet needs of people with dementia and their carers.

### Self-organisation in relational work and emergent 'social support' outcomes

Therapists engaged in building relationships with participants and their carers; the intervention handbook described one of the therapist's roles as '*Developing rapport with participants and carers and building good relationships' (Clare and Kudlicka, p42).*[27] In interviews, people with dementia described positive relationships with therapists:

Oh fine, yeah fine, got on well … Easy, yeah she explained everything and, you know, it was no hardship (laughs)…That's right, yeah, well sometimes when people come to see you, … you're afraid to talk, you know, afraid to say anything when it's a little bit dumb. But she made me feel so comfortable and within a couple of minutes we were just like as though we'd been friends for a long time.

(Interview, Person with dementia 1)

Therapists reported that relationships with participants were an important aspect of the intervention, helping to engage people with dementia in the intervention and motivate them:

Therapist 2: The actual nature of the relationship and that therapeutic rapport, which I think actually counts for a lot, but I think it largely goes unmeasured in a way, what we bring as people, and our relationship, and that's the motivating factor, and it's hard to know how you would measure that, but I think there is a lot about them getting to know you and you becoming part of their routine, and that goes a long way

Therapist 5: I think they open up don't they

(Therapist focus group)

A good relationship was also important for therapists' work in that getting to know participants well underpinned their ability to help participants identify relevant goals and develop personalised strategies.

Additionally, therapists recognised that their relational work and the provision of social support as being important in itself and would dedicate time specifically to this aspect of their visits:

> Therapist 6: I think sometimes it's that really good for the carer and the participant to have somebody going in and for such a long time. You do work on the goals, but another big part of it is quite, you know, supportive, and you know, social, because you spend the first ten minutes of your visit, they just go on about what they've done in the past week, and you can't be like, come on let's crack on [laughter] I think you do see in a lot of people, especially as you go on, they thaw a bit as you visit and then [several people agreeing] I get that good bond by session ten, and you're sad to pull out, but I think that support does really help.
>
> (Therapist focus group)

This relational work was self-organising in that therapists conducted relational work and provided social support, going slightly beyond the intervention parameters by regarding it as an intervention component in itself and spending more time on this than reflected in the handbook.

The social support provided by the therapists also developed into an emergent outcome of the intervention in that it was perceived as the main benefit for some participants:

> Therapist 7: the pleasure of seeing people try and do well, even if they're not achieving their goals, it's the other things that they're getting from it, the social interaction, the time to talk about their condition, the dementia, and it not being hushed away and in the cupboard.
>
> (Therapist focus group)

In interviews, some people with dementia were unable to recall the goals they had been working towards but many commented on the relationship with the therapist and that they would miss the visits now the therapy had ended. Carers also commented on relational benefits for people with dementia:

> I think my mum just enjoyed it more that somebody was, the social aspects of it, that somebody was coming
>
> (Interview, Carer 1)

The therapy log analysis indicated that relational and social support outcomes were an important (or at least the best-recalled) element of the intervention by people with greater cognitive impairment. At the end of the intervention, participants from the 'poor outcomes' group were more likely to refer to the relational or social aspect of the therapist visiting them as a positive element of the intervention, whereas participants in the 'good outcomes' group were more likely to give examples of formal components of the intervention that had benefited them. One therapist also commented in the focus group that social support could be the primary benefit for the carer.

In these ways, relational work conducted by therapists and their provision of social support extended the scope of intervention as it was described in the handbook. Self-organisation occurred through microlevel decisions of therapists which helped them deliver the intervention successfully but also meet the social support needs of participants and in doing so developed into an emergent outcome.

## DISCUSSION

This article provides an example of how process evaluation findings can be interpreted using complexity theory and the concept of self-organisation. Some implementation processes occurred as self-organisation, through individual decisions made by therapists. These decisions produced 'order' at a higher level by creating patterns of service delivery and outcomes. The self-organisation of the intervention occurred partly by design, through a person-centred approach, but also through adaptions (to severity of dementia), extensions (of therapists' time and social support) and add-on components (carer support and relationship conflict resolution).[34] The handbook recommended but did not include detail about the adaptation of the intervention, and also implied flexibility in relation to the delivery of formal components rather than extending the time allocation or scope of the intervention.[27] It was not always clear, therefore, where self-organisation fell within or outside of the intervention parameters. Some of the adaptations resulted from therapists' interpretation of the flexibility allowed within the intervention as an aspect of its personalised approach, to achieve participants' goals which was the core component of the intervention. Therapists are likely to have drawn on their professional training and experience to make decisions about when and how to adapt the intervention. For example, skilled therapists such as OTs would generally translate complex material for use with clients, such as simplifying language in a handbook. Therapists also felt they were going beyond the intervention design at times, particularly in feedback loops when they provided 'add-ons' in response to participants' needs and contexts such as addressing relationship conflicts. Implementation within a trial context meant the intervention was described in a structured way in the handbook and there was less flexibility than would be the case in normal clinical practice. This may have led therapists to view the intervention as relatively structured and been more likely to view their adaptations as falling outside the intervention parameters.

The findings about self-organisation have several implications for further studies or wider roll-out of the GREAT

intervention. First, the adaptations made by therapists for people with greater cognitive impairment (within the trial range of an MMSE score of ≥18 points) could be added to the handbook as examples of tailoring to the specific needs of the person, and inform training for therapists. Second, referrals to other services and specialised professionals could be enhanced in order to meet the wider needs of people with dementia and their carers, particularly for common issues such as relationship conflict. Third, social support needs could be addressed by referral to an alternative, less resource-intensive intervention such as befriending. Alternatively, the emergent outcome of social support, particularly for those with greater cognitive impairment, could be more formally incorporated into the intervention design. This may require increased resourcing, including formally incorporating this skill into recruitment and training for professionals, depending on what prior training practitioners have received.[35] This intervention was delivered mainly by skilled OT therapists who typically offer a range of types of support; if the intervention was delivered by OT assistants, for example, additional training might be required. This also depends on whether future interventions are defined as targeted CR interventions or expanded to incorporate the additional elements observed in this study. Fourth, the time allocation for visits could be revised, or made more flexible, to accommodate the additional time requirements for some visits due to the extended or 'add-on' components provided. Lastly, future evaluations of the intervention could investigate the impact of social support for participants and carers, since findings indicated this area was important but an in-depth examination of this was beyond the scope of the study.

This analysis had several limitations. First, the qualitative data are the views and perceptions of the therapists, participants and carers of how the intervention operated, and the therapy logs were relatively brief notes made by therapists. Second, only one focus group of six therapists was conducted, and the therapy log analysis was limited to 50 participants. However, this analysis was designed to produce theoretical generalisability through in-depth analysis rather than a large sample size.

Applying a complexity theory lens had advantages over traditional process evaluation approaches. It explained how decisions dispersed throughout a system at the microlevel interacted with population characteristics and context, and how this influenced what intervention was delivered and what outcomes it produced. It went beyond describing practitioner adaptations, as it showed how adaptation can create 'order' in general patterns of behaviour and in creating outcomes other than that which was originally intended by the intervention (in this case, social support). The idea of self-organisation also specifically conceptualises decisions made by practitioners as a 'bottom up' phenomenon which emphasises the less predictable nature of this kind of behaviour. This is in contrast to the fidelity and adaptation literature which emphasises defining core form and function,

and peripheral aspects of interventions, to identify what should be planned or controlled.[22 36 37] The types of self-organising feedback loops described here are likely to be a common pattern in health interventions, where practitioners adapt interventions to meet the needs of their clients. They may be particularly common where participants have high levels of unmet needs, comorbidities or in complex cases. Future studies of implementation could analyse the microprocesses of self-organisation specifically, in addition to formal components of interventions. This type of approach requires an agnostic position on fidelity of intervention delivery since lack of adherence to intervention protocols is not necessarily a negative aspect of the intervention.[38] Although some degree of fidelity is important for studies such as trials, understanding how adaptation may occur is important for real-world implementation. Adaptive behaviours could be positive in that they support individuals, help engage participants and help tailor the intervention to a local context.[14 39 40] However, adaptive behaviours could also create difficulties if they expand the scope of the intervention beyond what is possible for sustained delivery, or are unsuccessful. Expansion could displace the delivery of core components of the intervention, for example, or lead to burn-out in practitioners. This depends partly on how the intervention is being developed and refined during process evaluations: in some early-stage studies, adaptations by practitioners in practice may be helpful. However, when testing a well-defined intervention in a definitive trial, adaptations could be more problematic. Clarity about the balance required between fidelity and adaptation, and the trade-offs involved, is therefore necessary.[41]

A complexity perspective was also useful in that it identified informal, less visible processes which might not be picked up by standard evaluation models measuring fidelity against formal components and protocols. One particularly informal and non-visible aspect of the intervention was the relational work conducted by therapists and the provision of social support through this. Findings from this study were that it was an important aspect of how the intervention operated: it facilitated participant and carer engagement in the intervention, supported aspects such as personalisation, and underpinned improved social support valued by participants and their carers. While relational and social support work may be recognised by practitioners as a common microlevel process, it is given comparatively little attention in published health intervention research.[42 43] This study suggests it is worthy of more attention in health intervention evaluation studies.

Complexity theory was used here to retrospectively explain the implementation of an intervention, which is less useful than predictive theory. However, the latter is challenging.[10 44] Rather than being predictive in a strong sense, complexity theory could inform programme theories and logic models to include commonly-occurring patterns such as self-organisation or feedback loops.[45] Complexity constructs could be used deductively in qualitative data analysis, for example. In this study, they were

particularly useful in the later stages of analysis when findings were considered together to try to understand how the different elements of the intervention worked together. This could be supported through developing less linear logic models and applying them flexibly as programme theory develops during an evaluation.[3] [19] Emergent outcomes could be considered as one potential outcome in a logic model, in addition to pre-specified primary and secondary outcomes. Adaptation and feedback loops could also be incorporated in logic models, as suggested in the Medical Research Council (MRC) process evaluation guidance.[1] To this end, complexity theory could be more useful for process evaluation as a methodological theory to guide modelling, data collection and analysis, or an overarching framework, rather than a type of theory that explains how the world works in a strongly predictive sense.[9] [46] An alternative approach is to combine complexity approaches with other theoretical perspectives which explain agency or structures at different levels of a system.[6] [8] [47] Both approaches require multiple, exploratory and flexible methods, including qualitative methods of sufficient depth, in order to identify informal, complex and unpredictable patterns.[6] [7] [12] [40] [48] While complexity theory is increasingly employed, it is still a relatively new approach in complex health interventions research, particularly in empirical research.[40] [49] Calls have been made for further examples of its application as well as better operationalisation of its concepts.[1] [7] [49] [50] This article has provided one example of how complexity theory, particularly the concept of self-organisation, can be useful for providing insight into the implementation of an intervention that would have been missed by a process evaluation only focusing on formal intervention components.

**Author affiliations**
[1]College of Medicine and Health, University of Exeter, Exeter, UK
[2]School of Health and Social Work & NIHR Applied Research Collaboration East of England, University of Hertfordshire, Hatfield, UK
[3]School of Medicine and Global Brain Health Institute, Trinity College Dublin, Dublin, Ireland
[4]Centre for Applied Dementia Studies, University of Bradford, Bradford, UK
[5]Dementia Pal Ltd, QCS Quality Compliance Systems, Guildford, UK
[6]Dementia Services Development Centre Wales, School of Health Sciences, Bangor University, Bangor, UK
[7]College of Medicine and Health & NIHR Applied Research Collaboration South-West Peninsula, University of Exeter, Exeter, UK

**Acknowledgements** We thank all the participants with dementia and their family supporters who generously contributed their time to be part of this study. We also thank our experts by experience and staff at Alzheimer's Society for supporting the study. We would like to acknowledge the key roles played by Ho Yin Chan, Matthew Lewis and Julie Nixon who conducted the qualitative interviews, Dr Gill Toms who conducted some of the qualitative data analysis, and the trial therapists who provided the intervention.

**Contributors** SM-T (guarantor) designed and conducted the analysis of the focus group and therapy log data, oversaw the analysis and discussed themes from the interview data, and devised and drafted the paper; AK discussed themes developed from the focus group and therapy log data, and also commented on several drafts of the paper; KW analysed interview data and commented on several drafts of the paper; JRO conducted the focus group, contributed to the study design and commented on several drafts of the paper; IL, JP and RW contributed to the study design and commented on several drafts of the paper; LC oversaw the study design, discussed themes developed and commented on several drafts of the paper.

**Funding** This work was supported by NIHR HTA, grant number 11/15/04.

**Competing interests** None declared.

**Patient consent for publication** Not applicable.

**Ethics approval** The study was reviewed by Wales Rec 5, which issued a favourable opinion on 25 June 2012 (Reference 12/WA/0185), and was approved by the Bangor University School of Psychology Research Ethics Committee.

**Provenance and peer review** Not commissioned; externally peer reviewed.

**Data availability statement** No data are available. Qualitative interview and focus group data were not deposited in an archive due to the small numbers of individuals participating in named sites, which could compromise anonymity through potentially identifiable information in transcripts. Therapy logs data were clinical notes and therefore not appropriate for depositing in an archive.

**ORCID iDs**
Sarah Morgan-Trimmer http://orcid.org/0000-0001-5226-9595
Krystal Warmoth http://orcid.org/0000-0003-0615-5778

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
