## [Reviewer comments · BMJ Open]

ARTICLE DETAILS

TITLE (PROVISIONAL)	Implementation processes in a cognitive rehabilitation intervention for people with dementia: A complexity-informed qualitative analysis
AUTHORS	Morgan-Trimmer, Sarah; Kudlicka, Aleksandra; Warmoth, Krystal; Leroi, Iracema; Oyebode, Jan; Pool, Jackie; Woods, Robert; Clare, Linda

VERSION 1 – REVIEW

REVIEWER	Gates, Emily Boston College, Measurement, Evaluation, Statistics, & Assessment
REVIEW RETURNED	19-Apr-2021

GENERAL COMMENTS	This paper presents a clear, compelling example of using process evaluation informed by critical realist approach & complexity theory. Kudos for the clarity and focus in writing as it made the paper easy to understand and, I imagine, accessible to a wide audience. My minor suggestions are to make the specifics of the approach used even more clear so that others could apply this, if appropriate: 1. The paper shows the value of the approach taken as compared to implementation fidelity or a focus on implementation as designed and specified in the manual. It is clear that adaptation matters but less clear how these adaptation at the micro level led to a self-organizing pattern at the structure or whole-intervention level. What evidence is there that the patterns identified in the qualitative analyses of micro-level adaptations actually did occur at the whole-intervention level? If there is no evidence, what could be done to discern whether the individual adaptations were self-organizing patterns? Without this, it is hard to see what the concepts of feedback loops and self-organization add from complexity theory beyond conventional ideas of responsive tailoring and adaptation.2. How did the critical realist orientation inform the analysis? It would be helpful to have a bit more information about how the deductive codes were developed based on this framework or how this lens informed what was examined inductively. Again, for readers interested in applying a similar approach when studying implementation.
--

3. In the discussion, there is sufficient attention to how the results could inform tailoring of the intervention design and protocols. Given the study purpose to also test out the use of these complexity concepts and critical realist approach, it would be great to see some examples of what was learned from this study about how others could use self-organization & feedback loops as conceptual guides within qualitative process evaluations.

REVIEWER	Birken, SA
REVIEW RETURNED	Wake Forest School of Medicine 24-May-2021

GENERAL COMMENTS	Thank you for the opportunity to review bmjopen-2021-051255, "Implementation as self-organisation: findings of the GREAT trial process evaluation from a complexity perspective." The manuscript explores the applicability of the complexity perspective concept of self-organization to the implementation process. I appreciate the complexity perspective as it applies to healthcare delivery and implementation and welcome opportunities to consider new thinking about it. I have three four concerns that dampened my enthusiasm about this manuscript. Major First, I am concerned that the authors didn't acknowledge much of the literature that has applied a complexity lens in health services research. The journal Healthcare Management Review has published quite a bit on this, and it seems like a missed opportunity not to acknowledge that and work in other journals on the topic to emphasize the unique contribution of this particular manuscript. Otherwise, I worry that we're reinventing the wheel and not advancing our thinking in any appreciable way. The authors should do a deep dive on this literature and make very clear what their work adds conceptually given the large amount of extant work in this area. Perhaps the extant work applied to other clinical areas, but surely there's something to be learned from that work to accentuate the unique contribution of this manuscript. As currently written, it's unclear what the unique perspective is here. Second, I worry that the authors have conflated implementation with delivery. The authors at the outset of the paper distinguish their definition of implementation from that of the US National Institutes of Health; however, I can't subscribe to the authors' definition, which seems to me to be the definition of delivery, which is different from implementation. To be clear, my understanding is that delivery is one facet of an intervention – not implementation – and that the other facet of an intervention is its content. That is, there are likely facets of intervention delivery without which the intervention would be ineffective, as is the case for intervention content. From my perspective (and, I'd argue, from the perspective of many implementation researchers), implementation represents efforts to promote the proficient and consistent use of the intervention—the necessary components of its content and delivery. It was very difficult to read the paper and see the word implementation where I thought the word should
---

have been delivery. From my perspective, this is not just semantics; this misnomer has implications for delivery and for implementation.

Third, the manuscript doesn't seem to ground itself in the adaptation literature. The authors acknowledge self-organization as a kind of adaptation. The adaptation literature is small, but it's growing and gaining attention. Of particular note, Kirk et al. make the case that adaptation of an intervention (of relevance to this manuscript, its delivery) can render the intervention ineffective if the adapted components had been critical to its effectiveness. In this sense, the concept of self-organization is worrisome without a critical eye toward whether self-organization might in fact compromise the intervention's effectiveness. In fact, many of the results seem to point toward the idea that 'self-organization' in this case simply represents unplanned adaptation.

Fourth, the closest the authors seem to get to a research question seems to be: "This article examines the application of complexity concepts, particularly 'self-organisation', in a process evaluation and reflects on its use for understanding the implementation of complex health interventions." Still, this isn't precise enough for me to evaluate whether the methods are sufficiently rigorous enough to address this question and whether the results have answered the question. Related, the methods state, "A focus group and intervention therapy logs examined the perceptions and experiences of therapists, and a set of interviews explored the perceptions and experiences of participants and their carers." But perceptions and experiences as they relate to what? The applicability of the concept of self-organization? How was this assessed? And the authors do not define a critical realist perspective or how it was applied to the data. The authors identify themes relating to implementation on page 8, but they do not explain how they are relevant to implementation. It's also unclear how outcomes were measured (last paragraph before interviews section on page 8). Without a clear research question, it's very difficult to understand how these methods were used to accomplish the goal of the study. It also makes the discussion seem cherry-picked. "A complexity perspective was also useful in that it identified informal, less visible processes which might not be picked up by standard evaluation models measuring fidelity against formal components and protocols." This statement is very difficult to falsify given the lack of clear methods for addressing the research question, and in particular without a counterfactual evaluation model.

Minor

'successful' implementation; 'effective' confuses people with intervention effectiveness

REVIEWER	Dryden-Plamer, Karen University of Toronto, IHPME
REVIEW RETURNED	02-Jun-2021

GENERAL COMMENTS

Thank you very much for the opportunity to review this very robust process evaluation report. The reader can fully appreciate the hidden processes illuminated by applying the authors selected lens of complexity/self-organization to the phenomena. This is one of the strongest contributions from this work.

I have very few very minor queries.

The last line in the 1st paragraph of the abstract (line 10 page 2) is there a work missing .. cognitive rehabilitation programs/interventions?

The background might be further strengthened by identifying what about this program meets criteria as a complex intervention (line 39 page 3).

I thank the authors for describing the limitations of their design. 1/3 of the therapist were not participating in the focus group. Can the authors expand on how this might have impacted their data and results – were their differences between therapists who participate and those who did not?

Similarly, could the authors provide information on how the 50 therapy logs were selected-criteria for 'good outcomes' and 'poor outcomes' especially in light of goals and outcomes begin selected by the participants.

In the participants and care givers interviews it is not clear how service recipients were able to inform on the adaptation behaviours of the therapists- were participants provided with the program structure prior to receiving services so as to appreciate these micro adaptations? The interview schedule seems focused on perceptions of participant satisfaction with service and individual level program outcomes. Can the authors provide further explanation as to how this interdigitates with the self-organizational analysis?

The description of methods might also benefit by sharing the core concepts of complexity theory that guided the focus group analysis (line 18 page 7)

Appreciating the tension between intervention fidelity and 'fitting' the intervention it to the individual client context the authors have done an admirable job of describing the flexibility within the program to achieve a distinction between designed adaptation and to concept of 'self-organization'. It seems that the content of the intervention was the fixed element however the delivery of the intervention was open for adaptation (timing, extension of relationships, pace, language). The data shared in the report supported these conclusions very well.

It has been my pleasure to review this report and I thank you very much for sharing this interesting work.

VERSION 1 – AUTHOR RESPONSE

Reviewer: 1	
1. The paper shows the value of the approach taken as compared to implementation fidelity or a focus on implementation as designed and specified in the manual. It is clear that adaptation matters but less clear how these adaptation at the micro level led to a self-organizing pattern at the structure or whole-intervention level. What evidence is there that the patterns identified in the qualitative analyses of micro-level adaptations actually did occur at the whole-intervention level? If there is no evidence, what could be done to discern whether the individual adaptations were self-organizing patterns? Without this, it is hard to see what the concepts of feedback loops and self-organization add from complexity theory beyond conventional ideas of responsive tailoring and adaptation.	Evidence from the focus groups was with all of the therapists involved in the intervention at that time (additional therapists joined the study later on). We have clarified the text on page 7 to be clear that all therapists involved in the intervention at the time took part in the focus group. Agreement occurred between therapists (evidenced by more than one making the same point in quotes, or noting where agreement occurred in the group in the transcripts). Quotes are included from a range of therapists to show that they represent broad practice rather than the view of one or two individuals. Further, we drew from three datasets to build a comprehensive explanation of how the intervention operated and produced results. Together, these support our assertion about how the intervention operated according to complexity constructs. Qualitative methods tend to rely on theoretical generalisability rather than large sample sizes. We have generated a theoretical perspective based on a small setting, drawing on three data sources which supported our theoretical interpretation. We have presented the findings as potentially representing patterns which could be explored in further research, to avoid over-stating the findings. The complexity perspective is distinct from conventional ideas about adaptation because it identifies whole system patterns of behaviour and links them to outcomes. We have added comments in the discussion section, on page 14, to clarify why we think complexity theory is a useful conceptual approach over and above the traditional literature on implementation and adaptation: It went beyond describing practitioner adaptations, as it showed how adaptation can create 'order' in general patterns of behaviour and in creating outcomes other than that which was originally intended by the intervention (in this case, social support). The idea of self-organisation also specifically conceptualises decisions made by practitioners as a 'bottom up' phenomenon which emphasises the less predictable nature of this kind of behaviour. This is in contrast to the fidelity and adaptation literature which emphasises defining core form and function, and peripheral aspects of interventions, to identify what should be planned or controlled.
2. How did the critical realist orientation inform the analysis? It would be helpful to have a bit more information about how the deductive codes were developed based on this framework or how this lens informed what was examined inductively. Again, for readers interested in applying a similar approach	We have added the following paragraph to the beginning of the methods section on page 6 to explain how the critical realist approach shaped the data analysis: A critical realist perspective was adopted in the data analysis to identify causal patterns of how the intervention operated (a realist ontology) while allowing that perceptions of the intervention may differ between participants (a relativist epistemology). To this end, the three datasets were initially analysed with an overall focus on participant perspectives, whether these were categorised according to deductive categories identified from previous research or inductively developed. Additional interpretive work was conducted using a

when studying implementation.	graphic to relate themes from the focus group analysis to each other, to draw out a more realist explanation of the dynamic relationships between different elements of the intervention. In a second stage, a realist-oriented analysis of how the intervention operated was conducted though deductively applying constructs from the complexity literature tthe initial findings from the three datasets.
3. In the discussion, there is sufficient attention to how the results could inform tailoring of the intervention design and protocols. Given the study purpose to also test out the use of these complexity concepts and critical realist approach, it would be great to see some examples of what was learned from this study about how others could use self-organization & feedback loops as conceptual guides within qualitative process evaluations.	We have added some comments to the discussion section, on page 15, to add to our points about how others could use our approach in future research: Complexity constructs could be used deductively in qualitative data analysis, for example. In this study, they were particularly useful in the later stages of analysis when findings were considered together to try to understand how the different elements of the intervention worked together. [Existing text here: This could be supported through developing less linear logic models and applying them flexibly as programme theory develops during an evaluation (3, 16).] Emergent outcomes could be considered as one potential outcome in a logic model, in addition to pre-specified primary and secondary outcomes. Adaptation and feedback loops could also be incorporated in logic models, as suggested in the MRC process evaluation guidance.
Reviewer: 2	
First, I am concerned that the authors didn't acknowledge much of the literature that has applied a complexity lens in health services research. The journal Healthcare Management Review has published quite a bit on this, and it seems like a missed opportunity not to acknowledge that and work in other journals on the topic to emphasize the unique contribution of this particular manuscript. Otherwise, I worry that we're reinventing the wheel and not advancing our thinking in any appreciable way. The authors should do a deep dive on this literature and make very clear what their work adds conceptually given the large amount of extant work in this area. Perhaps the extant work applied to other clinical areas, but surely there's	We limited the literature review of complexity theory due to the journal's word limit, and referenced the articles most closely related to our area of healthcare research. We have also referenced papers (Greenhalgh 2018, Eppel and Rhodes 2018, Thompson et al 2016, Rusoja et al 2018) which summarise current thinking and research in the complexity literature. Health Care Management Review does not have many recent articles on complexity theory, and these also largely focus on hospitals in the US which has limited applicability to the setting we investigated in our research. However, we have added further references to the complexity literature, including one from Health Care Management Review. These are: Paley J, Eva G. Complexity theory as an approach to explanation in healthcare: A critical discussion. International Journal of Nursing Studies. 2011;48(2):269-79 Colón-Emeric CS, Lekan-Rutledge D, Utley-Smith Q, et al. Connection, regulation, and care plan innovation: a case study of four nursing homes. Health Care Manage Rev. 2006;31(4):337-46 Churruca K, Pomare C, Ellis LA, et al. The influence of complexity: a bibliometric analysis of complexity science in healthcare. BMJ Open. 2019;9(3):e027308

something to be learned from that work to accentuate the unique contribution of this manuscript. As currently written, it's unclear what the unique perspective is here.	Eppel EA, Rhodes ML. Complexity theory and public management: a 'becoming' field. Public Management Review. 2018;20(7):949-59 Rusoja E, Haynie D, Sievers J, et al. Thinking about complexity in health: A systematic review of the key systems thinking and complexity ideas in health. J Eval Clin Pract. 2018;24(3):600-6 Khan S, Vander Morris A, Shepherd J, et al. Embracing uncertainty, managing complexity: applying complexity thinking principles to transformation efforts in healthcare systems. BMC Health Services Research. 2018;18(1):192 Long KM, McDermott F, Meadows GN. Being pragmatic about healthcare complexity: our experiences applying complexity theory and pragmatism to health services research. BMC Med. 2018;16(1):94 Recent papers, including one by Trish Greenhalgh in 2018, have highlighted complexity theory as a fairly new approach in our field. This article is arguing for the application of complexity approaches in process evaluations of complex health interventions, providing an example, and does not make claims to make a major contribution to complexity theory more broadly. The contribution of our article, in this context, is summarised at the end of the paper on page 15: While complexity theory is increasingly employed, it is still a relatively new approach in complex health interventions research, particularly in empirical research. Calls have been made for further examples of its application as well as better operationalisation of its concepts. This article has provided one example of how complexity theory, particularly the concept of self-organisation, can be useful for providing insight into the implementation of an intervention that would have been missed by a process evaluation only focusing on formal intervention components. We have also slightly edited a sentence to indicate the wider application of complexity approaches in health research, on page 3: Complexity theory has been increasingly advocated as an alternative theoretical lens for understanding complex interventions, and is being increasingly applied across a range of healthcare research areas and evaluation studies.
Second, I worry that the authors have conflated implementation with delivery. The authors at the outset of the paper distinguish their definition of implementation from that of the US National Institutes of	Thank you for drawing our attention to this issue and for the comments which draw out distinctions in key concepts in the definitions of implementation. We have added text in the first paragraph of the introduction to articulate our definition of implementation more clearly and to explain our emphasis on delivery, in order to orient the reader to the focus of the analysis and findings:

Health; however, I can't subscribe to the authors' definition, which seems to me to be the definition of delivery, which is different from implementation. To be clear, my understanding is that delivery is one facet of an intervention - not implementation - and that the other facet of an intervention is its content. That is, there are likely facets of intervention delivery without which the intervention would be ineffective, as is the case for intervention content. From my perspective (and, I'd argue, from the perspective of many implementation researchers), implementation represents efforts to promote the proficient and consistent use of the intervention-the necessary components of its content and delivery. It was very difficult to read the paper and see the word implementation where I thought the word should have been delivery. From my perspective, this is not just semantics; this misnomer has implications for delivery and for implementation.	While there is overlap between the two definitions, process evaluations tend to focus on 'the quality and quantity of what is actually delivered during the evaluation' [reference to MRC process evaluation guidance added here] and it is this focus we adopt in this article. We would argue that there is a clear distinction between the 'implementation science' and 'process evaluation' literatures in how they define implementation (even though there are overlaps), and that we are using an accepted definition from our field. We are using and have referenced the MRC process evaluation guidance definition, which is widely accepted and highly cited in the health research literature. While this guidance acknowledges "the movement away from simply capturing what is delivered, towards understanding how implementation is achieved, and how interventions become part of the systems in which they are delivered" (Pg. 36), the definitions it provides for implementation retain a focus on intervention delivery during the evaluation: Page 8: Implementation – the process through which interventions are delivered, and what is delivered in practice. Key dimensions of implementation include: Implementation process – the structures, resources and mechanisms through which delivery is achieved; Fidelity – the consistency of what is implemented with the planned intervention; Adaptations – alterations made to an intervention in order to achieve better contextual fit; Dose – how much intervention is delivered; Reach – the extent to which a target audience comes into contact with the intervention Page 10: The term implementation is used within complex intervention literature to describe both post-evaluation scale-up (i.e. the 'development-evaluation-implementation' process) and intervention delivery during the evaluation period. Within this document, discussion of implementation relates primarily to the second of these definitions (i.e. the quality and quantity of what is actually delivered during the evaluation). Process evaluations necessarily have a narrower definition of implementation because they are often embedded in randomised controlled trials. The aim of a process evaluation is to answer basic questions such as whether an intervention is feasible and whether it can be delivered with fidelity. While process evaluations would address some contextual factors, examining processes of embedding an intervention within a health system are not productive to investigate before an intervention has been shown to be effective.
Third, the manuscript doesn't seem to ground itself in the adaptation literature. The authors acknowledge self-organization as a kind of adaptation. The adaptation	We have added references to the adaptation literature: Miller CJ, Wiltsey-Stirman S, Baumann AA. Iterative Decision-making for Evaluation of Adaptations (IDEA): A decision tree for balancing adaptation, fidelity, and intervention impact. J Community Psychol. 2020;48(4):1163-77

literature is small, but it's growing and gaining attention. Of particular note, Kirk et al. make the case that adaptation of an intervention (of relevance to this manuscript, its delivery) can render the intervention ineffective if the adapted components had been critical to its effectiveness. In this sense, the concept of self-organization is worrisome without a critical eye toward whether self-organization might in fact compromise the intervention's effectiveness. In fact, many of the results seem to point toward the idea that 'self-organization' in this case simply represents unplanned adaptation.

Kirk MA, Haines ER, Rokoske FS, et al. A case study of a theory-based method for identifying and reporting core functions and forms of evidence-based interventions. Transl Behav Med. 2019;11(1):21-33

Kirk AM. Adaptation. In: Nilsen P, Birken SA, editors. Handbook on Implementation Science. Cheltenham: Edward Elgar; 2020. p. 317-32

Movsisyan A, Arnold L, Copeland L, et al. Adapting evidence-informed population health interventions for new contexts: a scoping review of current practice. Health Res Policy Syst. 2021;19(1):13

We have added a comment about adaptation and a reference to Kirk on page 4:

The core functions related to ways in which the intervention activities produce mechanisms of action, and must be retained for an intervention to be effective, while the activities (form) of an intervention may be adapted.

We have added a comment in the discussion section on self-organisation and how it differs from the adaptation literature on page 14:

It went beyond describing practitioner adaptations, as it showed how adaptation can create 'order' in general patterns of behaviour and in creating outcomes other than that which was originally intended by the intervention (in this case, social support). The idea of self-organisation also specifically conceptualises decisions made by practitioners as a 'bottom up' phenomenon which emphasises the less predictable nature of this kind of behaviour. This is in contrast to the fidelity and adaptation literature which emphasises defining core form and function, and peripheral aspects of interventions, to identify what should be planned or controlled (22, 36).

We have slightly edited the section on page 14 where we discuss the positive and negative aspects of adaptation with respect to process evaluation, and added references to Kirk:

This type of approach requires an agnostic position on fidelity of intervention delivery since lack of adherence to intervention protocols is not necessarily a negative aspect of the intervention (37). Although some degree of fidelity is important for studies such as trials, understanding how adaptation may occur is important for real-world implementation. Adaptive behaviours could be positive in that they support individuals, help engage participants, and help tailor the intervention to a local context (14, 38, 39). However, adaptive behaviours could also create difficulties if they expand the scope of the intervention beyond what is possible for sustained delivery, or are unsuccessful. Expansion could displace the delivery of core components of the intervention, for example, or lead to burnout

	in practitioners. This depends partly on how the intervention is being developed and refined during process evaluations: in some early-stage studies, adaptations by practitioners in practice may be helpful. However, when testing a well-defined intervention in a definitive trial, adaptations could be more problematic. Clarity about the balance required between fidelity and adaptation, and the trade-offs involved, is therefore necessary.
Fourth, the closest the authors seem to get to a research question seems to be: "This article examines the application of complexity concepts, particularly 'self-organisation', in a process evaluation and reflects on its use for understanding the implementation of complex health interventions." Still, this isn't precise enough for me to evaluate whether the methods are sufficiently rigorous enough to address this question and whether the results have answered the question. Related, the methods state, "A focus group and intervention therapy logs examined the perceptions and experiences of therapists, and a set of interviews explored the perceptions and experiences of participants and their carers." But perceptions and experiences as they relate to what? The applicability of the concept of self-organization? How was this assessed? ...Without a clear research question, it's very difficult to understand how these methods were used to accomplish the goal of the study. It also makes the discussion seem cherry-picked. "A complexity perspective was also useful in that it identified informal, less visible processes which might not be picked up by standard evaluation models measuring fidelity against formal components and	We have clarified the research questions that the overall study, and the specific analyses reported in this paper, addressed on page 5: The paper draws on several datasets which answered a set of related questions for the process evaluation about how the intervention operated. The research questions were: how was the intervention delivered; what was the feasibility of delivering the intervention; was the intervention delivered with fidelity; what were the mechanisms of impact; what influenced treatment outcomes; what were participant experiences of the intervention; and what were therapists' experiences of delivering the intervention? The analysis reported here answered a further research question: how did the way the intervention was delivered influence whether and how participants benefitted? The broad aim of a process evaluation is to explain how an intervention operates to produce its outcomes, but process evaluations often only address different elements of an intervention separately. This approach, drawing on complexity concepts, was an attempt to conduct a more holistic analysis. We have clarified the text on page 6 to be more specific about the research questions addressed by each data collection method: A focus group and analysis of therapy logs from the intervention examined the perceptions and experiences of therapists about how the intervention was delivered, feasibility of the intervention, fidelity to the intervention protocol, and perceived factors affecting treatment outcomes. A set of interviews explored the perceptions and experiences of the intervention by participants and their carers, and whether and how any impact from the intervention was experienced by this group.

protocols." This statement is very difficult to falsify given the lack of clear methods for addressing the research question, and in particular without a counterfactual evaluation model.	
And the authors do not define a critical realist perspective or how it was applied to the data. ...	We have added detail about how the critical realist approached shaped the data analysis, on page 6: A critical realist perspective was adopted in the data analysis to identify causal patterns of how the intervention operated (a realist ontology) while allowing that perceptions of the intervention may differ between participants (a relativist epistemology). To this end, the three datasets were initially analysed with an overall focus on participant perspectives, whether these were categorised according to deductive categories identified from previous research or inductively developed. Additional interpretive work was conducted using a graphic to relate themes from the focus group analysis to each other, to draw out a more realist explanation of the dynamic relationships between different elements of the intervention. In a second stage, a realist-oriented analysis of how the intervention operated was conducted though deductively applying constructs from the complexity literature to the initial findings from the three datasets.
The authors identify themes relating to implementation on page 8, but they do not explain how they are relevant to implementation.	We identify themes from the initial focus group analysis, but do not fully explain them in the methods section as this information is more appropriately reported in the findings section. We have edited the description of the sentence describing the focus group themes in the methods section, on page 7, and also signposted that the themes are explained further in the findings section: This article draws on three themes relevant to implementation: the perceived influence of the severity of dementia; adaptation work of therapists in response to their perceptions of the severity of dementia; and the relational work implemented by therapists and the outcomes this produced. These themes are more fully described in the findings section.
It's also unclear how outcomes were measured (last paragraph before interviews section on page 8).	We have added detail about how the primary outcome was defined and measured, on page 7: Therapy logs were analysed to compare the 25 participants with the highest-score primary outcomes (the 'good outcomes' group) and the 25 participants with the lowest-score primary outcomes (the 'poor outcomes' group), out of the intervention arm population of 281 individuals. The primary outcome was participant-reported goal attainment at 3 months post randomisation, measured using the Bangor Goal-Setting Interview (BGSi). This measurement was undertaken through a home visit by a researcher who was blinded to the trial arm of the participant.

	We also added a reference to the HTA report which provides a description of the primary outcome and BGSi measure in more detail.
'successful' implementation; 'effective' confuses people with intervention effectiveness	We have changed 'effective' to 'successful' in the abstract and on pages 12 and 14.
Reviewer 3	
The last line in the 1st paragraph of the abstract (line 10 page 2) is there a work missing .. cognitive rehabilitation programs/interventions?	We have amended this sentence to “...which compared a cognitive rehabilitation intervention for people with dementia with usual treatment.”
The background might be further strengthened by identifying what about this program meets criteria as a complex intervention (line 39 page 3).	We have added a sentence to identify how the intervention is a complex intervention, on page 5: The intervention is conceptualised here as a complex intervention as it had multiple interacting components, addressed difficulties encountered in dementia for both participants and carers, was an individually tailored intervention and targeted multiple outcomes A reference to the MRC complex interventions guidance has also been added.
I thank the authors for describing the limitations of their design. 1/3 of the therapist were not participating in the focus group. Can the authors expand on how this might have impacted their data and results - were their differences between therapists who participate and those who did not?	We have checked study records and the focus group was with all of the therapists (six) involved in the intervention at that time (three additional therapists joined the study later on). We have edited the text on page 7 to correct this: A focus group was conducted with all six therapists who were in post at the time, at the end of the first year of the intervention, to examine their experiences of the intervention
Similarly, could the authors provide information on how the 50 therapy logs were selected-criteria for 'good outcomes' and 'poor outcomes' especially in light of goals and outcomes begin selected by the participants.	We have edited the following text to page 7 to explain how therapy logs were selected: Therapy logs were analysed to compare the 25 participants with the highest-score primary outcomes (the 'good outcomes' group) and the 25 participants with the lowest-score primary outcomes (the 'poor outcomes' group), out of the intervention arm population of 281 individuals. The primary outcome was participant-reported goal attainment at 3 months post randomisation, measured using the Bangor Goal-Setting Interview (BGSi). This was undertaken through a home visit by a researcher who was blinded to which trial arm the participant was in.

	We also added a reference to the HTA report which provides a description of the primary outcome and BGS1 measure in more detail. The role of participants in setting their own goals has also already been explained on page 5 in the section describing the trial: The intervention involved identifying with the participant personally relevant and significant goals related to daily activities, and then working together to develop and implement a set of strategies to enable the person to achieve the desired outcomes. The primary outcome was participant-reported progress towards participant-identified goals at three months.
In the participants and care givers interviews it is not clear how service recipients were able to inform on the adaptation behaviours of the therapists- were participants provided with the program structure prior to receiving services so as to appreciate these micro adaptations? The interview schedule seems focused on perceptions of participant satisfaction with service and individual level program outcomes. Can the authors provide further explanation as to how this interdigitates with the self-organizational analysis?	Interviewees were not asked about and did not report on adaptation behaviours of therapists. The purpose of the interviews was to examine participant experience and perceptions of the intervention. The link between their experiences and therapists' behaviours was made during the interpretive work of the analysis which drew findings together under the concepts of complexity theory to try to understand how the intervention worked as a complex system. Therapists' perceptions that they were delivering social support some of the time aligned with participant and carer reports that social support was one way in which they had benefitted from the intervention.
The description of methods might also benefit by sharing the core concepts of complexity theory that guided the focus group analysis (line 18 page 7)	We have added a list of complexity concepts drawn from the literature we cited, on page 8: In a last stage of analysis, themes from the focus group analysis were interpreted in relation to core concepts in the complexity literature. The core concepts considered were: interacting elements, unpredictability, self-organisation, emergence, non-linearity, fuzzy boundaries, feedback loops and being 'more than the sum of its parts' We also added additional references for this list of concepts.

VERSION 2 – REVIEW

REVIEWER	Gates, Emily Boston College, Measurement, Evaluation, Statistics, & Assessment
---

REVIEW RETURNED

17-Sep-2021

GENERAL COMMENTS

The article provides a clear example of using select complexity concepts within the qualitative analysis of data regarding the implementation of cognitive rehabilitation intervention. Using a complexity orientation instead of one focused on fidelity alone revealed adaptations made by carers. The fact that this analysis was embedded inside a randomized trial shows the benefits of blending approaches to research and including this kind of process evaluation within a larger experimental study. Prior comments were adequately addressed in this version. I have no further suggestions.